# Not Just an Alternative Energy Source: Diverse Biological Functions of Ketone Bodies and Relevance of HMGCS2 to Health and Disease

**DOI:** 10.3390/biom15040580

**Published:** 2025-04-14

**Authors:** Varshini V. Suresh, Sathish Sivaprakasam, Yangzom D. Bhutia, Puttur D. Prasad, Muthusamy Thangaraju, Vadivel Ganapathy

**Affiliations:** 1Department of Cell Biology and Biochemistry, Texas Tech University Health Sciences Center, Lubbock, TX 79430, USA; varshini.suresh@ttuhsc.edu (V.V.S.); sathish.sivaprakasam@ttuhsc.edu (S.S.); yangzom.d.bhutia@ttuhsc.edu (Y.D.B.); 2Department of Biochemistry and Molecular Biology, Augusta University, Augusta, GA 30912, USA; pprasad@augusta.edu (P.D.P.); mthangaraju@augusta.edu (M.T.)

**Keywords:** β-hydroxybutyrate, ketoacidosis, ketone body transporters, GPR109A, epigenetic modification, HMGCS2, loss-of-function mutations, inflammation, cancer, neurodegeneration

## Abstract

Ketogenesis, a mitochondrial metabolic pathway, occurs primarily in liver, but kidney, colon and retina are also capable of this pathway. It is activated during fasting and exercise, by “keto” diets, and in diabetes as well as during therapy with SGLT2 inhibitors. The principal ketone body is β-hydroxybutyrate, a widely recognized alternative energy source for extrahepatic tissues (brain, heart, muscle, and kidney) when blood glucose is sparse or when glucose transport/metabolism is impaired. Recent studies have identified new functions for β-hydroxybutyrate: it serves as an agonist for the G-protein-coupled receptor GPR109A and also works as an epigenetic modifier. Ketone bodies protect against inflammation, cancer, and neurodegeneration. HMGCS2, as the rate-limiting enzyme, controls ketogenesis. Its expression and activity are regulated by transcriptional and post-translational mechanisms with glucagon, insulin, and glucocorticoids as the principal participants. Loss-of-function mutations occur in HMGCS2 in humans, resulting in a severe metabolic disease. These patients typically present within a year after birth with metabolic acidosis, hypoketotic hypoglycemia, hepatomegaly, steatotic liver damage, hyperammonemia, and neurological complications. Nothing is known about the long-term consequences of this disease. This review provides an up-to-date summary of the biological functions of ketone bodies with a special focus on HMGCS2 in health and disease.

## 1. Introduction

Despite the tag “genesis”, ketogenesis is not an anabolic pathway; it is a catabolic pathway involving oxidative degradation (β-oxidation) of long-chain fatty acids and subsequent utilization of the end products of this oxidative process to generate ketone bodies. This process occurs predominantly in the liver, but other organs such as the small intestine, colon, kidney, retina, and brain are also capable of ketogenesis to a biologically significant extent.

Ketone bodies consist of acetoacetate, β-hydroxybutyrate, and acetone; acetoacetate is made first, which then gets converted to β-hydroxybutyrate. Acetone is produced from acetoacetate via nonenzymatic decarboxylation. The predominant ketone body in circulation is β-hydroxybutyrate. As these ketone bodies are water-soluble, they do not need lipoproteins or albumin for circulation; they exist in blood in free form. While acetoacetate and β-hydroxybutyrate exhibit varied biological functions, no biological role has yet been shown for acetone. Acetoacetate and β-hydroxybutyrate are oxidized in most organs, but not in the liver, for energy production and when present in excess, are excreted by the kidneys. Acetone, being volatile, is eliminated by the lungs. Under healthy and physiological conditions, the levels of ketone bodies in circulation are in the range of 0.05–0.4 mM, but these levels increase significantly (1–2 mM) during fasting as well as during strenuous exercise [1,2,3,4]. Overnight, 10–12 h after the previous meal, ketone bodies in blood begin to rise in the morning. Fasting for several days results in more than 20 times higher levels of ketone bodies than physiological levels (6–8 mM) [4]. Since acetoacetate (pKa, 4.0) and β-hydroxybutyrate (pKa, 4.4) are acids [5], excess levels of these metabolites cause ketoacidosis. This is commonly seen in uncontrolled type 1 diabetes but can also occur at a lower grade in type 2 diabetes (diabetic ketoacidosis), starvation for several days (starvation ketoacidosis), chronic use of alcohol (alcoholic ketoacidosis), and long-term use of ketogenic diets (nutritional ketoacidosis). Early symptoms of ketoacidosis include frequent urination, increased thirst, dry mouth, and headache; but when left untreated, the symptoms progress to include nausea and vomiting, fatigue, shortness of breath, mental issues such as confusion, and fruity-smelling breath (“keto breath”) due to acetone in the expired air.

## 2. Enzymatic Reactions Involved in Ketogenesis and Ketone Utilization

### 2.1. Ketogenesis

Both β-oxidation and ketogenesis occur within the mitochondrial matrix. The pathway involved in ketogenesis is depicted in Figure 1. Acetyl-CoA is the starting point for ketogenesis; acetyl-CoA arises mostly from β-oxidation of fatty acids but can also come from the catabolism of selective amino acids (ketogenic amino acids). First, two molecules of acetyl-CoA condense to form acetoacetyl-CoA by the reaction catalyzed by thiolase, also known as acetyl-CoA acetyltransferase 1 (ACAT1) or β-ketothiolase [6]. Acetoacetyl-CoA then combines with another molecule of acetyl-CoA to form 3-hydroxy-3-methylglutaryl-CoA (HMG-CoA), a reaction catalyzed by HMG-CoA synthase-2 (HMGCS2). HMG-CoA is then broken down to acetoacetate and acetyl-CoA by the HMG-CoA lyase (HMGCL), and the resultant acetoacetate is then reduced to β-hydroxybutyrate by β-hydroxybutyrate dehydrogenase-1 (BDH1) using NADH as the electron donor. β-Hydroxybutyrate is a chiral molecule because of the asymmetry of carbon-3 due to the presence of the hydroxyl group and therefore can exist either as an R (also known as the D-form) or S (also known as the L-form) enantiomer [7]. BDH1 generates only the R enantiomer; thus, the physiological form of this ketone body is R-β-hydroxybutyrate.

The amino acids that provide carbon for ketogenesis (i.e., ketogenic amino acids) are either purely ketogenic (leucine and lysine) or both ketogenic and glucogenic (isoleucine, phenylalanine, tyrosine, tryptophan, and threonine). Among them, leucine and isoleucine deserve special mention due to some unique aspects related to their metabolism. Catabolism of leucine involves the generation of HMG-CoA as an intermediate without the participation of HMGCS2 [8,9]. Catabolism of isoleucine involves ACAT1, the same enzyme that participates in ketogenesis and ketone utilization (see below); as a result, inactivating mutations in ACAT1 lead to disruption of not only the metabolism of ketone bodies but also the metabolism of isoleucine [10].

### 2.2. Ketone Utilization

Extrahepatic tissues utilize ketone bodies as an energy source in lieu of glucose when glucose availability is limited or glucose metabolism is impaired. This utilization pathway (i.e., ketolysis) also occurs within the mitochondrial matrix; here β-hydroxybutyrate is converted to acetyl-CoA (Figure 1), which then enters the citric acid cycle to generate the reducing equivalents NADH and FADH_2_ as the electron donors for the electron transport chain. But the conversion of β-hydroxybutyrate to acetyl-CoA is not simply the reversal of the ketogenesis pathway [11]. However, BDH1 and β-ketothiolase (ACAT1) participate in both processes (BDH1: acetoacetate to β-hydroxybutyrate in ketogenesis and β-hydroxybutyrate to acetoacetate in ketone utilization; ACAT1: acetyl-CoA to acetoacetyl-CoA in ketogenesis and acetoacetyl-CoA to acetyl-CoA in ketone utilization). As such, BDH1 and ACAT1 mediate their reactions in the direction of ketogenesis in the liver while the same enzymes work in the direction of ketolysis in extrahepatic tissues. What differs between ketogenesis and ketolysis is the mechanism for the reversible conversion between acetoacetyl-CoA and acetoacetate. In ketogenesis, acetoacetyl-CoA is converted to acetoacetate with the involvement of two enzymes, namely HMGCS2 and HMGCL, with HMG-CoA as an intermediate, whereas in ketolysis, acetoacetate is converted to acetoacetyl-CoA with the involvement of a single enzyme known as succinyl-CoA:3-ketoacid CoA transferase-1 (SCOT1) without the production of HMG-CoA as an intermediate [12]. SCOT1 is also known as 3-oxoacid CoA transferase-1 (OXCT1).

## 3. Ketogenesis in Different Organs

### 3.1. Liver

Liver is the most predominant site for ketogenesis, serving as the primary source of ketone bodies in circulation [1,2,3]. This metabolic pathway, along with the β-oxidation of fatty acids that supply acetyl-CoA for ketogenesis, is accelerated; however, it is accelerated only when the levels of glucose in circulation drop, as in starvation, or glucose metabolism is impaired, as in diabetes. Such physiological or pathological conditions are also associated with accelerated gluconeogenesis. Glucose is the primary source of metabolic energy for the brain, and when glucose availability or utilization is restricted, lipolysis occurs in adipocytes to release fatty acids and glycerol into circulation. The liver takes up the fatty acids to generate the ketone bodies and at the same time uses glycerol as the carbon source for gluconeogenesis. Lactate and glucogenic amino acids also serve as additional carbon sources for the synthesis of glucose. The resultant ketone bodies serve as an effective alternative source of metabolic energy for the brain. The integration of lipolysis in adipocytes, and β-oxidation/ketogenesis in association with gluconeogenesis in the liver is made possible because of the increase in circulating levels of glucagon relative to insulin under these conditions [13]. With this hormonal milieu, cAMP/protein kinase-A-dependent cellular signaling becomes active in adipocytes and hepatocytes because of the activation of the glucagon receptor on the cell surface, which underlies many of the metabolic changes that occur in these two tissues.

It is important to note that the hormonal milieu (i.e., increased glucagon/insulin ratio) that is conducive to simultaneously promoting ketogenesis and gluconeogenesis does not involve solely the enhanced cAMP signaling. The decreased function of insulin also plays an active role because insulin effectively antagonizes the cAMP/protein kinase-A-mediated protein phosphorylation cascade by activating protein phosphatase-1 [14].

When the restriction of glucose availability is extended for a long time, as in prolonged starvation, the stress hormone cortisol comes into play as an additional promoter of gluconeogenesis by supplying glucogenic amino acids to the liver via increased protein breakdown in the skeletal muscle. When this happens, ureagenesis is revved up in the liver to handle the nitrogen from the amino acids that is released in the form of ammonia when the carbon skeleton of these amino acids is used for gluconeogenesis.

Acetyl-CoA generated in β-oxidation not only serves as the substrate for ketogenesis to produce ketone bodies and for the citric acid cycle to produce ATP but also functions as a promoter of gluconeogenesis. Pyruvate is an intermediate when lactate and several glucogenic amino acids are used in gluconeogenesis; this pyruvate is preferentially channeled to form oxaloacetate by pyruvate carboxylase instead of going through the reaction catalyzed by pyruvate dehydrogenase complex to form acetyl-CoA. This happens because excess acetyl-CoA resulting from β-oxidation inhibits the pyruvate dehydrogenase complex and at the same time activates pyruvate carboxylase. Oxaloacetate then goes through the reaction mediated by phosphoenolpyruvate carboxykinase to produce phosphoenolpyruvate, which then follows the rest of the gluconeogenic pathway to generate glucose. As such, there is effective coordination among the four different metabolic pathways in the liver, namely β-oxidation, ketogenesis, gluconeogenesis, and ureagenesis (i.e., urea cycle), all occurring simultaneously [13].

While the reduced levels of insulin and the concomitant increased levels of glucagon have received the most attention as a result of the coordinated occurrence of β-oxidation, ketogenesis, gluconeogenesis, and urea cycle in the liver, recent studies have shown that FGF21 (fibroblast growth factor 21), secreted primarily by the liver, elicits metabolic effects similar to those of glucagon in the liver [15,16]. The mechanism of action for FGF21 is, however, different from that of glucagon. The cell-surface receptor for FGF21 is a complex consisting of FGFR1 and β-klotho and the downstream signaling inside the cells involves mitogen-activated protein kinase (MAPK), extracellular signal-regulated kinases 1 and 2 (ERK1/2), as well as AMP-activated protein kinase (AMPK) [16].

The hormonal and biochemical mechanisms that operate in diabetic ketoacidosis and starvation ketoacidosis are similar to a large extent (Figure 2). In diabetes, insulin is either low (type 1 diabetes) or hypofunctional due to resistance (type 2 diabetes) and glucagon secretion is higher in both cases, thus resulting in an increased glucagon/insulin ratio in terms of their biological action. A similar situation exists during starvation. Because of reduced glucose availability from diet, blood levels of glucose decrease in starvation, which decreases insulin secretion and increases glucagon secretion. FGF21 is also secreted at higher levels in the liver in diabetes as well as in starvation. Consequent to these changes in the hormonal milieu, ketogenesis is revved up markedly in the liver with excessive production of the ketone bodies, thus leading to ketoacidosis.

Alcoholic ketoacidosis is primarily due to alcohol-induced hypoglycemia, increased availability of acetate arising from alcohol metabolism, and alcohol use-associated reduced caloric intake (Figure 2). Liver is the principal site of alcohol metabolism, and the two enzymes involved in the catabolic pathway, namely alcohol dehydrogenase and aldehyde dehydrogenase, generate NADH from NAD^+^, and also acetate as the end product. The resultant increase in the NADH/NAD^+^ ratio in liver cells prevents the entry of oxaloacetate, pyruvate, and dihydroxyacetone phosphate into gluconeogenesis via the promotion of the conversion of these intermediates into malate, lactate, and α-glycerophosphate, respectively. As such, chronic use of alcohol interferes with gluconeogenesis and causes hypoglycemia. This would decrease insulin secretion and increase glucagon secretion, a situation similar to that seen in diabetes and starvation. In addition, acetate arising from alcohol gets converted to acetyl-CoA to feed into ketogenesis.

The mechanisms underlying nutritional ketoacidosis are similar to those in starvation ketoacidosis to a significant extent (Figure 2). Ketogenic diets are characterized by low carbohydrate and high fat content. The reduced availability of glucose from diet tends to decrease the circulating levels of glucose, thus increasing the glucagon/insulin ratio in blood. In addition, the high fat content in these diets increases the availability of fatty acids for β-oxidation. This hormonal and biochemical scenario promotes ketogenesis and causes nutritional ketoacidosis.

### 3.2. Kidney

The earliest study describing the kidney as a ketogenic organ dates back more than five decades, in which the ability of rat kidney cortical slices to convert the fatty acid oleate to ketone bodies was demonstrated [17]. Since then, several studies have substantiated in vivo the function of the kidney as a ketogenic organ in mice, at least during starvation [18,19] and in diabetes [20], and in rabbits with the infusion of acetate as a carbon source for ketogenesis [21]. This remains true in humans too; kidneys are capable of ketogenesis during starvation to a significant extent [22]. It is of interest to note that the expression of HMGCS2, the rate-limiting enzyme in ketogenesis, is comparable at the protein level in humans between the liver and kidney as assessed by immunohistochemical analysis even though mRNA levels are several-fold higher in the liver [23]. Despite all these findings that support the ketogenic function of the kidney, it remains debatable whether this organ actually contributes to circulating levels of ketone bodies to any physiologically relevant extent [19,24]. Speculation has been put forth that the kidneys are indeed capable of ketone body production, but these ketone bodies may elicit local functions within the kidney as an energy source as well as in intracellular signaling rather than getting secreted into blood [19]. Additional studies are needed in this area focusing not only on physiological conditions but also on pathological states such as long-term starvation and liver diseases.

### 3.3. Small Intestine and Colon

Publicly available databases show significant expression of mRNA for HMGCS2 in the human small intestine, colon, and rectum, though much less than that found in the liver, but HMGCS2 protein levels are comparable to those in the liver [23]. The presence of HMGCS2 in the human small intestine [25,26] and colon [26] has also been demonstrated independently by immunolocalization and Western blot. However, whether this is true in other species remains controversial. Some studies have found no detectable Hmgcs2 protein in the mouse small intestine even though the presence of the protein was detectable in the colon [27,28], but others have shown convincingly the presence of Hmgcs2 protein in organoids generated from mouse small intestinal crypts [26,29,30]. The reasons for this discrepancy may include differences in the age and dietary conditions of the animals used in the studies. There is strong evidence for robust expression of Hmgcs2 in rat intestine during suckling and subsequent decrease to undetectable levels in adults [31,32,33]. It is therefore likely that the ketogenic enzyme is not expressed in the adult small intestine in rodents. In these animals, ketogenesis in the small intestine is a function that occurs only during the early stages of development, restricted only to the suckling period. The expression of the enzyme in mouse small intestinal organoids might be due to the specific conditions used in the organoid culture system involving intestinal crypts and multiple growth factors. Directly relevant to this issue is the demonstration of Hmgcs2 expression almost exclusively in stem cells in the crypts in mice, both in the small intestine [29,34,35] and colon [27]. This is in contrast to the expression pattern in the human intestine and colon, where the protein is expressed more robustly in differentiated surface epithelial cells than in crypt cells [26,36]. In adult mice, the protein expression for Hmgcs2 is comparable between the cecum and liver, but significantly less so in the colon [28], indicating potential involvement of microbiota and bacterial metabolites in the control of Hmgcs2 expression in the host. This is due to the unique feature of cecum in this animal species as the major reservoir for feces, and hence for microbiota and bacterial metabolites.

A recent study has assessed the specific contribution of ketogenesis in the intestinal tract to blood ketones in mice with the deletion of Hmgcs2 in the entire intestinal tract (small intestine as well as large intestine) using Villin-Cre in Hmgcs2-floxed mice [28]. A significant decrease in ketone bodies in circulation was found in these mice even when hepatic ketogenesis was intact, thus demonstrating at least some contribution of intestinal tract ketogenesis to blood ketones, though considerably less than that of the liver.

### 3.4. Retina

There is evidence for ketogenesis in retinal pigment epithelial cells [37,38]. The evidence includes HMGCS2 mRNA and protein expression as well as the ability of the cells to synthesize and release ketone bodies when incubated with the fatty acid palmitate. These cells are polarized, with their basolateral membrane facing choroidal blood and the apical membrane facing the inner retina and in contact with photoreceptor cells. It has been shown that the release of ketone bodies occurs across the apical membrane to provide energy substrates for photoreceptor cells, which express the enzymatic machinery to utilize ketone bodies for energy production. A more recent study [39], however, has shown that the release does not occur preferentially across the apical membrane but occurs across apical as well as basolateral membranes. Interestingly, phagocytosis of the outer segments of photoreceptor cells induces ketogenesis in retinal pigment epithelial cells, hence leading to the speculation that the fatty acids present in these outer segments in the form of phospholipids serve as the starting material for β-oxidation and subsequent ketogenesis [40].

### 3.5. Astrocytes

Astrocytes in the brain provide metabolic support to neurons that is essential for the maintenance of efficient and optimal neurotransmission. This includes the synthesis and release of glutamine to neurons for use as the immediate precursor of the neurotransmitter glutamate (glutamine–glutamate cycle) and the supply of glucose-derived lactate to neurons as an important energy source. In a similar vein, astrocytes also metabolize fatty acids and generate ketone bodies that are released into the extracellular space for neuronal use as an additional alternative energy substrate [41]. Astrocytes in culture do synthesize ketone bodies from fatty acid oxidation [42]. It appears that the capacity of astrocytes for ketogenesis is magnified when glucose availability to the brain is decreased, as evidenced by the activation of ketone body production in astrocytes by hypoxia and hypoglycemia [43]; AMP kinase, a cellular marker for low energy status [44]; long-term starvation [45]; and conditions of impaired glycolysis [46]. Thus, astrocytes serve as metabolic sensors and reprogram their metabolism to provide alternative fuel molecules to neurons, which include not only lactate but also ketone bodies, under stressful and adverse conditions [47].

## 4. Transporters for Ketone Bodies

Acetoacetate and β-hydroxybutyrate, generated within the mitochondrial matrix, get across the inner mitochondrial membrane to enter the cytoplasm, but there is no information available in the literature on the identity of the transporter(s) responsible for this process. But we do know which transporters are likely responsible for the transport of these molecules across the plasma membrane. Acetoacetate and β-hydroxybutyrate exist under physiological conditions predominantly in the form of ionized monovalent anionic monocarboxylates. As such, these ketone bodies need specific transporters to traverse the plasma membrane during release from the cells or uptake into cells. Because of the chemical nature of these molecules, monocarboxylate transporters provide the sole mechanism for this process. There are two distinct classes of monocarboxylate transporters, Na^+^-coupled and H^+^-coupled, belonging to two different solute carrier (SLC) gene families, with the Na^+^-coupled transporters in the SLC5 family and the H^+^-coupled transporters in the SLC16 family [48,49,50,51]. Among them, the transport of acetoacetate and/or β-hydroxybutyrate has been demonstrated for SLC5A8 (SMCT1), SLC16A1 (MCT1), SLC16A3 (MCT4), SLC16A7 (MCT2), SLC16A8 (MCT3), and SLC16A6 (MCT7). The highest affinity for β-hydroxybutyrate is for SLC5A8 and SLC16A7 (*K*_t,_ ~1 mM). Whether a given transporter functions in the influx or efflux of ketone bodies is dictated by the coupling of its transport process to Na^+^ or H^+^, the concentration gradients of the ketone bodies across the plasma membrane, and its substrate affinity. Sometimes, the same transporter may work in influx or efflux depending on the cell type as to whether the cells synthesize or use the ketone bodies.

### 4.1. Influx Transporters

SLC5A8 (SMCT1, sodium-coupled monocarboxylate transporter 1) is an influx transporter; it transports a variety of monocarboxylates such as lactate, acetoacetate, β-hydroxybutyrate, pyruvate, and short-chain fatty acids (acetate, propionate, and butyrate), in addition to the B-complex vitamin nicotinate [52,53,54,55]. The transport process is energized by the transmembrane Na^+^ gradient and is electrogenic with a Na^+^: monocarboxylate stoichiometry of 2:1, thus highly suitable to function as an influx transporter. The transporter is localized in the lumen-facing apical membrane of colonocytes and renal proximal tubular epithelial cells, basolateral membrane of the retinal pigment epithelium, and neurons [49]. It is therefore likely that SLC5A8 contributes to the reabsorption of ketone bodies from the glomerular filtrate, transfer of ketone bodies from the choroidal blood into neural retina, and influx of ketone bodies into neurons from extracellular milieu.

The H^+^-coupled monocarboxylate transporter SLC16A1 (MCT1, monocarboxylate transporter 1) is highly expressed in endothelial cells that form the blood–brain barrier [56] and inner blood–retinal barrier [57], where it is responsible as an influx transporter for the transfer of ketone bodies in circulation into the brain and retina. The same transporter also functions in the uptake of ketone bodies in oxidative cancer cells [58]. In retinal pigment epithelium, SLC16A8 (MCT3) functions as the influx transporter at the basolateral membrane [59], where it works in conjunction with SLC5A8 for the cellular uptake of circulating ketone bodies from the choroidal blood. Neurons express SLC16A7 (MCT2), which exhibits the highest affinity for ketone bodies [60]; as such, neurons employ two transporters, namely SLC5A8 and SLC16A7, to take up ketone bodies from the extracellular space.

### 4.2. Efflux Transporters

Efflux transporters are needed not only for the release of ketone bodies into the extracellular space from cells that generate them by ketogenesis but also for the transcellular transfer of ketone bodies in polarized cells such as the renal tubular cells, endothelial cells of the blood–brain barrier and inner blood–retinal barrier, and retinal pigment epithelium. In most cases, SLC16A1 (MCT1) performs this function. This transporter is present in the basolateral membrane of colonocytes and renal tubular cells, abluminal membrane of the barrier-associated endothelial cells, and apical membrane of retinal pigment epithelium. As such, SLC16A1 serves as an influx transporter in some cell types and as an efflux transporter in others. It is an electroneutral transporter with a H^+^: monocarboxylate stoichiometry of 1:1 and in most cases, there is no significant transmembrane H^+^ gradient to energize this transporter in any given direction. Therefore, the concentration gradient for the ketone bodies across a given membrane dictates the direction of transport.

More recently, SLC16A6 (MCT7) has been identified as a selective transporter for the efflux of ketone bodies [61]. Absence of this transporter in zebra fish (red moon mutant) increases the levels of triglycerides in the liver, suggesting that the inability of liver cells to export ketone bodies results in the suppression of fatty acid oxidation and consequent accumulation of triglycerides. The phenotype of hepatic steatosis seen in these mutants is reversed when genetically forced to express either wildtype zebra fish Slc16a6 or human SLC16A6, demonstrating the functional similarity between zebra fish and human transporters. The actual transport function of human SLC16A6 as a selective transporter for the ketone body β-hydroxybutyrate was established by heterologous expression of the transporter in *X. laevis* oocytes. In SLC16A6-expressing oocytes, exposure to β-hydroxybutyrate, but not to lactate, induced inward currents under voltage-clamped conditions. No other details are available on the transporter function, particularly regarding the mechanism underlying the inward current and the nature of coupled cations (H^+^, Na^+^, or others). Nonetheless, the function of SLC16A6 as a selective transporter for the release of ketone bodies from cells received supporting evidence from a recent study in which downregulation of SLC16A6 in the human colon cancer cell line Caco-2 decreased the extracellular levels of β-hydroxybutyrate without affecting the levels of lactate or pyruvate [62]. In contrast, the downregulation of SLC16A1 (MCT1) or SLC16A3 (MCT4) led to a decrease in the efflux of lactate and pyruvate. These findings show that, unlike other MCTs that can recognize lactate as well as β-hydroxybutyrate as substrates, SLC16A6 (MCT7) selectively accepts β-hydroxybutyrate as its substrate. Interestingly, this transporter also appears to accept taurine as a substrate, but the transporter-mediated uptake of taurine is competitively inhibitable by β-hydroxybutyrate [63]. It is important to note that the expression of SLC16A6 is widespread in human tissues (cerebellum, adrenal gland, salivary gland, small intestine, colon, liver, kidney, testis, epididymis, ovary, fallopian tube, and heart) [64] and that this distribution pattern does not match that of the ketogenesis enzyme HMGCS2 (stomach, small intestine, colon, liver, gallbladder, kidney, urinary bladder, testis, and mammary gland) [23]. SLC16A6 (MCT7) may not be a transporter exclusively for β-hydroxybutyrate and is likely to have other endogenous substrates as well.

## 5. Biological Functions of Ketone Bodies

### 5.1. Alternative Energy Substrate During Limited Glucose Availability

Mammalian tissues have characteristic preference for metabolic fuels to generate energy that is tissue-specific, primarily based on the enzymatic machinery expressed in a given tissue, which is dictated not only by the expression of the specific genes but also by the presence or absence of specific subcellular organelles. For example, mature erythrocytes use glucose exclusively; because of the absence of mitochondria, no other metabolic pathways participate in this cell to generate ATP. For other cell types that possess mitochondria and hence the ability to use a variety of metabolic fuels, the order of preference is not the same; it varies from tissue to tissue: glucose > fatty acids for liver; glucose > ketone bodies for brain; glucose > fatty acids > ketone bodies for kidney and skeletal muscle; fatty acids > glucose > ketone bodies for heart. Based on this order of fuel preference, two key points emerge: (i) the liver, which is the predominant ketogenic organ, does not use ketone bodies as an energy source whereas all other major organs do; and (ii) for all other organs that are capable of using ketone bodies, the order of preference is such that ketone bodies come to the rescue only when the availability of other metabolic fuels, particularly glucose, becomes limited.

As detailed in Figure 1, the tissues that use ketone bodies for energy production employ the following pathway: β-hydroxybutyrate → acetoacetate → acetoacetyl-CoA → acetyl-CoA. Acetyl-CoA then enters the citric acid cycle for energy production. The entire pathway occurs within the mitochondria. The second step, which is unique for ketone utilization, requires the enzyme succinyl-CoA:3-ketoacid CoA transferase 1 (SCOT1 or OXCT1). This is also the rate-limiting step in the pathway. This enzyme is expressed in the brain, skeletal muscle, heart, and kidney, the tissues capable of using ketone bodies for energy production. The biological function of ketone bodies as a fuel assumes paramount importance in the brain (and retina) because neurons can use only glucose or ketone bodies for energy; therefore, when glucose availability is limited, as in prolonged starvation or genetic diseases associated with defective transfer of glucose into the brain (e.g., mutations in the glucose transporter GLUT1/SLC2A1), ketone bodies in circulation become the only other energy source for the brain (and retina). This is also true in uncontrolled type 1 diabetes when glucose utilization is impaired due to insulin deficiency despite the presence of plentiful glucose in circulation (starvation in the midst of plenty!). Accordingly, ketone bodies and ketogenic diets have become clinically useful for the treatment of several neurological diseases that may be associated with energy deficiency and also to preserve organ function in the brain, kidney, and heart in severe diabetes as well as in other pathological conditions. There have been numerous detailed, up-to-date reviews in recent years on this topic [65,66,67,68,69]. Diabetic ketoacidosis is mostly regarded with a negative connotation because of the clinical sequelae caused by severe metabolic acidosis in this condition, but the beneficial impact of the high levels of ketone bodies in the preservation of organ function in the brain, retina, kidney, and heart should not be ignored.

Since OXCT1 is the critical rate-limiting enzyme in ketone utilization, some more recent discoveries about this enzyme deserve mention. First, the mitochondrial protein frataxin, which is involved in the biogenesis of Fe-sulfur clusters, interacts with OXCT1 and protects the latter from ubiquitin-proteasomal degradation [70]. Consequently, Freidreich’s ataxia, which arises due to a genetic deficiency of frataxin, is associated with decreased levels of OXCT1, and hence defective ketone utilization and resultant ketoacidosis. Second, OXCT1 not only functions in ketolysis as a CoA transferase from succinyl-CoA to convert acetoacetate to acetoacetyl-CoA (Figure 1) but also functions as a succinyltransferase from succinyl-CoA to other proteins [71]. Third, the catalytic function of OXCT1 itself is regulated by post-translational modification involving succinylation [72]. Succinyl-CoA ligase 2 (SUCLA2), a subunit of the citric acid cycle enzyme succinyl-CoA synthetase, binds to OXCT1 and succinylates, the latter using succinyl-CoA with resultant activation of the catalytic function of OXCT1. It is interesting to note that these three recent findings center around succinyl-CoA, which is not only the substrate for OXCT1-mediated ketolysis and succinylation but also a promoter of the catalytic function of OXCT1 via post-translational succinylation. Furthermore, succinyl-CoA is the starting substrate, along with glycine, in heme biosynthesis. Since Fe is common in the formation of heme and iron–sulfur clusters, the involvement of frataxin in the control of OXCT1 protein stability is also of some unique interest and may have hitherto unidentified biological significance.

### 5.2. Hormone-like Signaling Functions via Cell-Surface G-Protein-Coupled Receptors (GPRs)

The most exciting discovery with regard to ketone bodies in terms of significance, next only to their well-known function as an alternative energy substrate, is their ability to elicit intracellular signaling by interacting with specific G-protein-coupled receptors on the cell surface [73,74,75]. GPR109A (originally identified as PUMA-G) was the first receptor identified to interact with the ketone body β-hydroxybutyrate with resultant activation of receptor signaling [76]. This receptor is G_i_/G_o_-coupled, and its activation by β-hydroxybutyrate inhibits adenylate cyclase (coupling with G_i_) and decreases the levels of cAMP inside the cells; in some cells, the result of GPR109A activation is an increase in intracellular Ca^2+^ (G_o_-coupled). Interestingly, GPR109A was first identified as the receptor for the B-complex vitamin niacin, but considering the normal plasma levels of this vitamin (~0.05 μM) and the concentration needed to activate the receptor (*EC*_50_, ~0.1 μM), it was soon realized that niacin is just a pharmacological agonist. This prompted the search for the physiological agonist, which led to the discovery of β-hydroxybutyrate as the endogenous agonist. Even though normal plasma levels of β-hydroxybutyrate under fed conditions (0.05–0.4 mM) are significantly less than the *EC*_50_ for GPR109A activation (~0.8 mM), the circulating levels of β-hydroxybutyrate do rise to 1–2 mM during fasting and vigorous exercise, and to even higher levels during prolonged starvation and uncontrolled diabetes that are sufficient to activate the receptor. It is important to note that acetoacetate, the other ketone body in circulation, does not interact with this receptor [76]. Interestingly, β-hydroxybutyrate is the physiological agonist for GPR109A only in non-colonic tissues. The receptor is expressed in colonic epithelial cells but on the lumen-facing apical membrane where the receptor has no access to β-hydroxybutyrate in blood. Instead of this ketone body, the bacterial fermentation product butyrate serves as the physiological agonist for the receptor at this site [77]. Even though the *EC*_50_ for activation of GPR109A by butyrate is ~1.6 mM [76], the concentration of this bacterial metabolite in colonic lumen is about 10 mM, which makes butyrate a physiologically relevant agonist for the receptor.

Some studies have shown that β-hydroxybutyrate functions as an agonist for another G-protein (G_q_ or G_i/o_)-coupled receptor, GPR41 (also known as free fatty acid receptor 3, FFA3) [78], but others have found the same ketone body to function as an antagonist for this receptor [79]. The reasons for this discrepancy remain unexplored.

Acetoacetate, though present at lower concentrations in blood than β-hydroxybutyrate, also functions as a signaling molecule. It is an agonist for the G-protein-coupled receptor GPR43 (also known as FFA2), which is also coupled to either G_q_ or G_i/o_ [80]. In HEK293 cells expressing GPR43, acetoacetate decreases forskolin-induced increase in cAMP in a dose-dependent manner (*EC*_50_, 0.76 mM), clearly demonstrating its agonist activity with G_i_ coupling. In adipocytes that express GPR43, acetoacetate increases intracellular levels of Ca^2+^ and activates the ERK signaling cascade. β-Hydroxybutyrate does not interact with this receptor. A decrease in cAMP has also been demonstrated in neutrophils downstream of GPR43 activation by acetoacetate [81]. In addition, the role of acetoacetate in muscle regeneration has been described in one study [82], but the receptor responsible for this function was not identified. In fact, the authors speculated the presence of an intracellular binding protein for acetoacetate as the likely mediator of the effects observed in this tissue. A more recent study has also demonstrated acetoacetate as a ligand for GPR43 in the brain in vivo [83], but surprisingly, the signaling pathways markedly differ. The interaction of acetoacetate with GPR43 in this study results in an increase in cAMP and a decrease in the ERK signaling cascade. These results could be interpreted as an antagonistic activity of acetoacetate on the receptor in the presence of some endogenous agonist under in vivo conditions. Obviously, additional research is needed to resolve these issues.

As can be inferred based on the widespread expression of GPR109A, GPR41, and GPR43 and their coupling to multiple G proteins (G_q_, G_i_, and G_o_) with associated diverse intracellular signaling, the biological consequences of ketone bodies as signaling molecules are pleiotropic. These receptors are expressed in adipocytes, immune cells, bone cells, endothelial cells, and adult stem cells, to name a few. There are excellent reviews published on this topic which detail these broad biological functions and their relevance to health and disease [7,24,66,67,68,73,74,84,85,86].

### 5.3. Epigenetic Modulation via Inhibition of Class-I/IIa Histone Deacetylases

β-Hydroxybutyrate, but not acetoacetate, has the ability to modulate epigenetic control of gene expression as well as the acetylation status and hence the function of specific non-histone proteins via inhibition of class I histone deacetylases (HDAC1-3) and class-IIa histone deacetylase HDAC4 [87,88,89,90,91,92,93,94,95]. However, even though some studies have been able to demonstrate HDAC inhibition by β-hydroxybutyrate at physiologically relevant concentrations (1–5 mM) [87,92,93], other studies found significant inhibition only at concentrations higher than 10 mM. There are also reports that β-hydroxybutyrate does not have any effect on HDACs even at 10 mM [96]. This discrepancy could be due to tissue-specific expression of HDAC isoforms because only class I/IIa HDACs show significant inhibition with β-hydroxybutyrate. HDACs control gene expression by regulating the acetylation status of histones, particularly lysine residues 9 and 14 in histone 3. The target proteins that are affected by β-hydroxybutyrate-mediated inhibition of HDACs include FOXO3a, catalase, mitochondrial superoxide dismutase (SOD2) [87,91], thioredoxin-1 [93], p53 [95], claudin-5 [92], brain-derived neurotropic factor [88], *N*-methyl-D-aspartate receptor [94], and the glucose transporter GLUT1 (SLC2A1) [89]. Some of these proteins are affected by HDACs via epigenetic control of their gene expression, whereas others (e.g., thioredoxin-1, p53, c-Myc, and NF-κB) are HDAC targets at the protein level, which controls their acetylation status and hence their biological function (e.g., p53) or proteasomal degradation (e.g., thioredoxin-1). As such, the biological processes impacted by the β-hydroxybutyrate/HDAC axis are broad, encompassing antioxidant machinery (FOXO3a, SOD2, catalase, thioredoxin-1), microvascular integrity (claudin-5), glucose transport into the brain (SLC2A1), glutamatergic neurotransmission (NMDA receptor), the cell cycle, DNA repair and apoptosis (p53), inflammation (NF-κB), and cancer (c-Myc). The impact of β-hydroxybutyrate on the acetylation status of histones and non-histone proteins is not only caused by HDAC inhibition but also by increased availability of acetyl-CoA, derived from β-hydroxybutyrate metabolism, as a substrate for acetyltransferases.

### 5.4. Post-Translational Modification via β-Hydroxybutyrylation

Covalent modification of lysine residues in histones and some non-histone proteins with β-hydroxybutyrate, known as β-hydroxybutyrylation, represents one of the most recently discovered post-translational modifications [97,98]. Similar to the impact of histone acetylation, β-hydroxybutyrylation of histones also weakens the interaction between histones and DNA and thus allows transcription factors to bind to promoters of target genes and facilitate transcription. The first report on this novel biological function of β-hydroxybutyrate was by Xie et al. [99], which showed β-hydroxybutyrylation of lysine in histones and consequent changes in transcription of specific genes in diabetic liver tissue. Histones that are subject to this modification include H2A, H2B, H3, and H4, but modification at K9 of H3 occurs most predominantly [99,100]. The first step in β-hydroxybutyrylation is the synthesis of β-hydroxybutyryl-CoA from β-hydroxybutyrate to serve as the substrate for the modification. This is most likely catalyzed by acyl-CoA synthetase short-chain family member 2 (ACSS2), which is well known for its ability to convert acetate, propionate, and butyrate into their respective CoA derivatives. Two enzymes participate in the transfer of the β-hydroxybutyryl group from β-hydroxybutyryl-CoA to histones: the acetyltransferases p300 and CREB-binding protein [100]. The reverse reaction, which removes the β-hydroxybutyryl group from the modified histones, is mediated by histone deacetylases HDAC1 and HDAC2 and by NAD^+^-dependent deacetylases SIRT3, SIRT5, and SIRT7 [99]. The genes whose transcription is promoted by β-hydroxybutyrylation of histones include BDNF, VEGF, metalloproteinase-2, adiponectin, and the ketogenic enzyme HMGCS2 (reviewed in [97]). Several non-histone proteins are also direct targets for β-hydroxybutyrylation. This includes the tumor suppressor p53, the citric acid cycle enzyme citrate synthase, and the methylation-related enzyme S-adenosyl homocysteine hydrolase; the biological functions of all three proteins are enhanced by their modification via β-hydroxybutyrylation (also reviewed in [97]).

Since the same sites in histones are involved in modification by acetylation and β-hydroxybutyrylation, a question arises as to whether the consequences of these two distinct modifications in terms of transcription of target genes are the same or different. The rationale for this question is that both acetylation and β-hydroxybutyrylation of lysine residues in histones remove the positive charge on the side-chain of the amino acid, thus weakening the interaction between histones and DNA and consequently opening up the promoter regions of the target genes to facilitate the binding of appropriate transcription factors and other regulatory proteins. Furthermore, as HDACs are inhibited by β-hydroxybutyrate, some of the effects on gene transcription observed with β-hydroxybutyrate could, in theory, be mediated by increased acetylation of histones rather than by β-hydroxybutyrylation. Based on a recent report, it appears that the consequences of acetylation and β-hydroxybutyrylation are not the same [101]. In blood–brain barrier endothelial cells, β-hydroxybutyrate increases the transcription of the tight-junction protein ZO-1, an effect that is mediated by increased β-hydroxybutyrylation of K9 of histone H3 at the promoter of the gene coding for the protein. However, inhibition of HDACs does not have any effect on ZO-1 expression. One possible explanation for the differences in transcriptional effects of acetylation versus β-hydroxybutyrylation may be that acetylated H3K9 and β-hydroxybutyrylated H3K9 are recognized by different transcriptional regulatory proteins (the so-called “readers”) in a modification-specific manner. Additional research is warranted in this area to tease out the relationship between histone lysine modifications by acetylation and β-hydroxybutyrylation in terms of biological function.

## 6. 3-Hydroxy-3-Methylglutaryl-CoA Synthase-2 (HMGCS2)

### 6.1. HMGCS2 Versus HMGCS1

Ketogenesis occurs within the mitochondrial matrix, and the rate-limiting enzyme in this pathway is 3-hydroxy-3-methylglutaryl-CoA synthase-2 (HMGCS2). This enzyme catalyzes the condensation of acetoacetyl-CoA and acetyl-CoA to produce 3-hydroxy-3-methylglutaryl-CoA (HMG-CoA). HMG-CoA is also an intermediate in the synthesis of cholesterol, but this pathway occurs predominantly in cytoplasm. As the CoA derivative of HMG cannot cross the mitochondrial membrane, a separate enzyme in the cytoplasm catalyzes the generation of HMG-CoA for cholesterol production. This enzyme, called HMG-CoA synthase-1 (HMGCS1), also catalyzes the condensation of acetoacetyl-CoA and acetyl-CoA and generates HMG-CoA. As such, even though the two isoforms of HMGCS mediate the same chemical reaction, the presence of two different enzymes with differential compartmentalization within the cell is necessitated because of the inability of HMG-CoA, the common intermediate in ketogenesis and cholesterologenesis, to freely traverse the mitochondrial membrane. In addition to the different subcellular localization between HMGCS2 and HMGCS1, their tissue distribution is also markedly different. As ketogenesis occurs only in certain selective tissues (e.g., liver, kidney, intestinal tract) whereas most tissues are capable of cholesterol synthesis, it is reflected in the tissue distribution of the two enzymes (Table 1) [23,102,103]. 

Within mitochondria, HMG-CoA generated by HMGCS2 is cleaved by HMG-CoA lyase to produce acetoacetate and acetyl-CoA. In cytoplasm, HMG-CoA generated by HMGCS1 undergoes NADH-dependent reduction by HMG-CoA reductase to produce mevalonate, which then gets into the isoprenoid pathway to produce cholesterol as the principal product. Additional products of this isoprenoid pathway include ubiquinone, dolichol, and farnesyl groups. Despite this seemingly clean separation between HMGCS1 and HMGCS2 in terms of their biological functions, there is evidence of significant functional overlap between the two enzymes. HMGCS2 is expressed in gonadal tissues (testis and ovary) where there is no evidence of ketogenesis [104]. The protein localizes to Leydig cells in the testis and theca cells in the ovary, and these cells may use HMGCS2 to synthesize cholesterol within mitochondria for utilization in steroid biosynthesis [105]. Concordant with the conclusion that Leydig cells do not generate ketone bodies but produce cholesterol within mitochondria is the absence of the ketogenic enzyme HMG-CoA lyase and the presence of the cholesterologenic enzyme HMG-CoA reductase in the inner mitochondrial membrane.

The role of HMGCS2 in cholesterol synthesis is further supported by the effective functional complementation in the HMGCS1-deficient Chinese hamster ovary cell line Mev-1 by ectopic expression of HMGCS2 [106]. Normally, Mev-1 cells do not survive unless the culture medium contains mevalonate because the mevalonate pathway is absent in these cells due to the lack of HMGCS1. But the cells are effectively rescued when engineered to express HMGCS2 and now the cells grow without addition of mevalonate to the medium. Synthesis of cholesterol within mitochondria using HMGCS2-generated HMG-CoA may be just one of the explanations for this functional complementation. It is also possible that mitochondrial HMG-CoA gets hydrolyzed to HMG, which passes through the inner mitochondrial membrane via a yet unknown mechanism to reach the cytoplasm, where it gets converted back into its CoA derivative to feed into the mevalonate pathway to generate cholesterol. The converse also seems to be true, at least in some cancer cells, where HMGCS1 mediates ketogenesis in collaboration with HMG-CoA lyase [107]. Apparently, HMG-CoA lyase is not exclusively a mitochondrial enzyme; in some cancer cells, HMGCS1 and HMG-CoA lyase colocalize and together they generate acetoacetate in the cytoplasm. BDH2 present in the cytoplasm reduces acetoacetate into β-hydroxybutyrate. HMG-CoA lyase activity has been detected in the cytoplasmic compartment in normal cells also [108,109], suggesting that HMG-CoA generated by the action of HMGCS1 in the cytoplasm could serve as the precursor for generation of ketone bodies outside the mitochondria even in normal cells. The cytosolic HMG-CoA lyase is the product of a gene that is distinct from the one that encodes the mitochondrial isoform [109]. Another confounding factor in the biological overlap between the two enzymes is that both mediate the same chemical reaction using acetyl-CoA and acetoacetyl-CoA. Therefore, it is possible to envision some kind of competition between the two enzymes for the same substrates. Even though the subcellular location of the reactions mediated by the two enzymes are distinct, acetyl-CoA can readily get across the inner mitochondrial membrane in the form of citrate following citrate synthase action within the mitochondria and subsequent lysis of citrate in the cytoplasm by ATP-citrate lyase to generate acetyl-CoA. Because of this competition, deficiency of one enzyme may increase the availability of acetyl-CoA for the other enzyme. This is supported by a recent study in which loss of HMGCS2 leads to increased synthesis of cholesterol [110].

### 6.2. Human HMGCS2: Gene, Protein, and Catalytic Mechanism

HMGCS2 was first cloned from rat liver [111] and subsequently from human liver [112]. The lead author of these cloning studies has published two outstanding reviews describing the general structural and catalytic features as well as transcriptional regulation of HMGCS2 [113,114]. The human HMGCS2 protein consists of 508 amino acids with a molecular weight of ~55 kDa. The gene encoding the protein is located on chromosome 1p12-13; it consists of ten exons, with exon 1 coding for the mitochondrial targeting sequence [115]. The promoter region of the enzyme possesses cis-elements for transcriptional regulation by peroxisome proliferator-activated receptor PPARα, hepatocyte nuclear factor-4, and cAMP-responsive element binding protein. In addition, the insulin regulatory element and glucocorticoid regulatory element are also present. The functional enzyme is a homodimer. The catalytic active site with its 21-amino-acid-long sequence corresponds to the region 151–171 (DSGNTDIEGIDTTNACYGGTA) in the human protein, but the site is one-hundred percent conserved in all mammals. The thiol group of the cysteine residue in this sequence is responsible for the covalent binding of the acetyl group with the release of free CoA in the initial step, followed by condensation of acetoacetyl-CoA with the enzyme-bound acetyl group to form HMG-CoA, still bound to the cysteine residue. As a final step, the product is released from the active site by hydrolysis. This is the classical ping-pong mechanism of enzyme catalysis.

### 6.3. Small-Molecule Inhibitors of HMGCS2

Hymeglusin, a fungal β-lactone with multiple names (antibiotic 1233A, antibiotic F-244, or compound L-659,699), is the most widely used small-molecule inhibitor of HMG-CoA synthase. It was first discovered to be an inhibitor of HMGCS1 involved in cholesterol synthesis in the cytoplasm [116,117,118]. It is a highly potent (*IC*_50_, ~0.1 μM) and irreversible inhibitor of HMGCS1, and the irreversibility arises from the covalent binding of the inhibitor via a thioester linkage to the active-site cysteine. Since the 21-amino acid active site is identical for HMGCS1 and HMGCS2, hymeglusin is also used as an effective inhibitor of the latter enzyme. A synthetic β-lactone, identified as DU-6622, also inhibits these enzymes with similar efficacy and an identical mechanism [119]. A recent study reported computational in silico screening data for interaction of HMGCS2 with several FDA-approved drugs sourced from the DrugBank database [120]; the study identified the top ten drugs that showed the highest binding affinity with theoretically calculated values for an inhibition constant of less than 0.1 μM. The list included rolapitant, penfluridol, rubitecan, lumacaftor, eltrombopag, estradiol cypionate, revaprazan, risperidone, ziprasidone, and lurasidone. Among these, penfluridol and lurasidone were found to form stable protein–ligand complexes with HMGCS2. However, these findings are solely based on theoretical analysis, and whether or not these drugs interact with HMGCS2 and inhibit its enzymatic activity has not been investigated through direct experimentation.

### 6.4. Regulation of HMGCS2 Expression and Activity

The expression and catalytic activity of HMGCS2 and therefore ketogenesis are under a complex regulatory control that involves both transcriptional (Figure 3) as well as post-translational (Figure 4) mechanisms (comprehensively reviewed in [113,114,121,122,123,124,125]). Several hormones participate in this process with profound metabolic implications for physiological conditions such as fed and starvation states and in pathological conditions such as diabetes, cancer, inflammation, and neurodegeneration. Insulin and glucagon are the primary players in HMGCS2 regulation, but other hormones such as glucocorticoids, catecholamines, thyroid hormone, and fibroblast growth factor FGF21 also contribute to the process. As mentioned at the beginning of this review, insulin and glucagon are counter-regulatory, and therefore the relative levels of these two hormones in circulation determine the status of HMGCS2 expression and activity. The insulin/glucagon ratio is higher during a fed state and the ratio reverses during starvation. Insulin suppresses the expression and catalytic activity of HMGCS2, whereas glucagon does the opposite.

#### 6.4.1. Regulation at the Level of Transcription of the HMGCS2 Gene

The transcriptional control of HMGCS2 expression by insulin and glucagon occurs via at least three pathways, all involving the respective hormone receptors on the plasma membrane of target cells (Figure 3). The promoter of the HMGCS2 gene contains cis-elements for binding of the transcription factors CREB (cyclic AMP-responsive element binding protein) and FOXA2 (forkhead box protein A2). Both these transcription factors activate HMGCS2 gene transcription. Insulin, primarily a fed-state hormone, suppresses HMGCS2 transcription; in contrast, glucagon, which is primarily a starvation-state hormone, enhances HMGCS2 transcription. Insulin activates phosphatidylinositol-3-kinase (PI3K) with subsequent activation of the downstream protein kinase B (PKB/AKT), a serine/threonine kinase [126]. PKB/AKT then phosphorylates FOXA2, which prevents the nuclear action of FOXA2 on the HMGCS2 promoter. On the other hand, glucagon acts through cAMP/protein kinase A, which phosphorylates CREB. The phosphorylated CREB directly binds to CRE (cAMP-responsive cis-elements) on the HMGCS2 promoter to activate transcription. The phospho-CREB also binds to lysine-acetyltransferase CBP (CREB-binding protein), and the resultant complex, together with another lysine-acetyltransferase, p300, acetylates FOXA2 and promotes its nuclear translocation with resultant activation of the HMGCS2 promoter [127]. Here too insulin plays a negative role in suppressing HMGCS2 expression by inducing the deacetylase SIRT1, which antagonizes the action of CBP/p300 on FOXA2. CBP/p300 also acetylates histones associated with the HMGCS2 promoter and facilitates transcription [127].

The HMGCS2 promoter also contains a cis-element sequence for binding to the nuclear receptor PPARα (peroxisome proliferator-activated receptor α), a very prominent participant in inducing HMGCS2 transcription [121,122]. PPARα works in partnership with another nuclear receptor, RXR (retinoid X receptor), along with other coactivators. Free fatty acids and fatty acid derivatives such as leukotrienes are the endogenous ligands, and fibrates (e.g., fenofibrate, gemfibrozil) are the pharmacological ligands for PPARα. Glucocorticoids (mainly cortisol), which are secreted in response to stress during long-term starvation, act through their nuclear receptor to promote PPARα expression. This requires the coactivator PGC1α (peroxisome proliferator-activated receptor gamma coactivator 1α). Since PGC1α also complexes with other nuclear receptors such as PXR (pregnane X receptor) and CAR (constitutive androstane receptor), PXR and CAR interfere with the action of PPARα on the HMGCS2 promoter by competing with PPARα for PGC1α. FGF21, which induces PGC1α expression, potentiates the transcriptional activity of PPARα on the HMGCS2 promoter [121,128]. PPARα is induced by another nuclear receptor, AhR (aryl hydrocarbon receptor) [129]. This finding is of biological importance in the liver and colon for HMGCS2 expression because these organs are often exposed to environmental xenobiotics, many of which activate AhR. Furthermore, colonic bacteria generate a variety of tryptophan metabolites (i.e., indole derivatives) and these metabolites are potent activators of AhR [130]. We have recently reported that the expression of Hmgcs2 in the colon is decreased in germ-free mice [28], indicating that colonic bacteria play an obligatory role in maintaining optimal expression of this enzyme in the colon. Short-chain fatty acids (acetate, propionate, and butyrate), generated in high quantities (the combined concentration of all three fatty acids can be as high as 80 mM in colonic lumen) by bacterial fermentation of undigested carbohydrates and dietary fiber, serve as the substrates for ketogenesis in the colon [131]. Thus, normal bacteria in the colon have a dual role in ketogenesis at this site: one role is to maintain high expression of the rate-limiting enzyme and the other role is to provide the necessary starting materials for the ketogenic pathway. An interesting feature of HMGCS2 regulation by PPARα is the feed-forward impact of HMGCS2 on the transcriptional activity of the receptor [132,133]. HMGCS2 directly complexes with PPARα, and the complex translocates into the nucleus; the efficiency of PPARα with respect to inducing the HMGCS2 promoter is significantly enhanced by this interaction with HMGCS2.

Another signaling pathway related to the transcriptional regulation of HMGCS2 expression is mediated by Wnt/β-catenin (reviewed in [122]). This pathway controls the expression of the transcription factor c-Myc, which is a repressor of HMGCS2 transcription. As such, activation of the Wnt/β-catenin signaling pathway reduces HMGCS2 expression. A recent report on this area is of interest because of the implication of an amino acid transporter in the control of this pathway [134]. SLC38A4 is a Na^+^-coupled transporter for neutral as well as cationic amino acids and is expressed almost exclusively in the liver, muscle, and placenta [135,136]. This transporter increases the levels of AXIN1, a protein that promotes the destruction of β-catenin [134]. As such, the amino acid transporter suppresses Wnt/β-catenin signaling and hence promotes HMGCS2 transcription with c-Myc as an intermediate.

Another recent finding that has profound clinical significance is related to the antidiabetic drugs known as gliflozins (e.g., empagliflozin), which block the renal reabsorption of glucose by inhibiting kidney-specific sodium-coupled glucose transporter 2 (SGLT2/SLC5A2) with nanomolar affinity. This class of drugs cause mild ketoacidosis in diabetic patients with a significant elevation of ketone bodies in circulation [137,138]. It is believed that this rather unexpected side effect may underlie the protective effects of these drugs in the kidney and heart by providing alternative energy substrates [139]. The ketoacidosis associated with these drugs involves the induction of HMGCS2 and the underlying molecular mechanism may involve the activation of AMP-activated kinase (AMPK) (Figure 2 and Figure 3) [140,141]. Decreased glucose entry into cells because of the blockade of SGLT2/SLC5A2 leads to a decrease in ATP production in the kidney, thus increasing the AMP/ATP ratio and the resultant activation of AMPK. AMPK negatively controls mTOR, which in turn negatively controls PPARα expression. The net result of this class of anti-diabetic drugs is to increase PPARα levels, thus enhancing HMGCS2 transcription. However, it is not clear whether the inhibition of SGLT2/SLC5A2 is the only mechanism for the ketoacidosis as the induction of HMGCS2 expression in response to these drugs is seen in tissues that do not express SGLT2/SLC5A2, suggesting that there may be additional pharmacological targets for gliflozins that are connected to HMGCS2 transcription [142].

#### 6.4.2. Regulation at the Level of Post-Translational Modification of HMGCS2 Protein

The catalytic activity of HMGCS2 is regulated by post-translational modification of the protein by phosphorylation, acetylation, and succinylation (Figure 4) [7,143,144,145,146,147,148]. Glucagon-regulated cAMP-dependent protein kinase A phosphorylates HMGCS2, and the result of this phosphorylation is an increase in the catalytic activity of the enzyme. Insulin, on the other hand, promotes removal of the phosphate group via protein phosphatase 1, thus antagonizing the effect of glucagon. In contrast to the functional consequence of phosphorylation, succinylation of HMGCS2, possibly catalyzed by OXCT1, reduces its catalytic activity. This regulation requires succinyl-CoA, the cellular levels of which are reciprocally controlled by glucagon (reduction) and insulin (elevation). The mitochondrial SIRT5 functions as a desuccinylase to reverse the succinylation. Acetylation of HMGCS2 also leads to decreased catalytic activity. Here acetylation is mediated by the lysine-acetyltransferases CBP and p300 and the reverse reaction is mediated by HDACs 1 and 2 and also by SIRT3.

## 7. HMGCS2 in Pathological Conditions

### 7.1. Diabetes

Diabetes is a disease associated with an increase in the glucagon/insulin ratio in terms of the biological activity of these two hormones, and hence with an increase in cellular levels of cAMP and a decrease in cellular levels of succinyl-CoA. As discussed in the previous section, these changes will increase the transcription of the HMGCS2 gene and also increase the catalytic activity of the protein, thus leading to enhanced ketogenesis. The same changes in the relative levels of these two hormones also fuel the substrate availability for ketogenesis by promoting lipolysis in adipocytes and increasing the circulating levels of free fatty acids, coupled with the promotion of β-oxidation of fatty acids in the liver.

### 7.2. Inflammation

Chronic inflammation is a hallmark of a wide variety of pathological conditions including diabetes, atherosclerosis, obesity, and neurodegenerative diseases. Ketone bodies have been shown to be effective in suppressing inflammation by blocking the NOD-like receptor family pyrin domain containing 3 (NLRP3) inflammasome [149,150]. This effect is mostly mediated by the agonistic action of the ketone body β-hydroxybutyrate on GPR109A. Several studies have documented the immunosuppressive role of GPR109A [151,152]. Accordingly, the rate-limiting enzyme in ketogenesis, HMGCS2, functions as an immunosuppressor. In the colon, HMGCS2 expression is substantially decreased in inflammatory bowel diseases (Crohn’s disease, ulcerative colitis) [26,28,153]. Moreover, ketogenesis protects against experimentally induced colonic inflammation [26,28,153]. This protective action is the result of site-specific local effects of ketogenesis in the colon without the participation of hepatic ketogenesis because conditional deletion of Hmgcs2 specifically in the intestinal tract (the small intestine as well as the colon) exacerbates experimental colitis in mice [28]. Since increased levels of glucose in circulation as in diabetes promote inflammation and contribute to the etiology of dysfunction in multiple organs such as the heart, kidney, and retina, ketone bodies are likely to provide beneficial effects in protecting organ function in such pathological conditions. HMGCS2 being the key enzyme in ketogenesis is likely to play a central role in these processes.

### 7.3. Cancer

The involvement of ketogenesis, and hence HMGCS2, in cancer is complex because of the multiple biological functions of the ketone bodies that may be either tumor-promotive or tumor-suppressive. Ketone bodies could function as an energy source for cancer cells in addition to glucose and amino acids and hence promote tumor growth. In contrast, the principal ketone body β-hydroxybutyrate could function as a signaling molecule to activate GPR109A and inhibit HDACs, both of which could suppress tumor growth. Available studies in published literature have demonstrated convincingly the tumor-promoting role of ketone bodies in breast cancer; the underlying mechanism is the utilization of these molecules as energy substrates by tumor cells [58,154]. Interestingly, this involves local production of ketone bodies by tumor-associated fibroblasts and the SLC16A1 (MCT1)-mediated influx of the ketone bodies in the tumor microenvironment into tumor cells for use in energy production. Breast cancer cells themselves also express HMGCS2, where the enzyme is associated with resistance to endocrine therapy with tamoxifen, but the mechanism underlying the link between HMGCS2 and tamoxifen resistance seems to be related to changes in mitochondrial oxidative stress [155].

In most other cancers, however, HMGCS2 functions as a tumor suppressor. This has been demonstrated in liver cancer, colorectal cancer, and prostate cancer. Most of the published research concerns cancer of the liver, the organ that normally has the most robust HMGCS2 expression. As ketogenesis is always associated with increased fatty acid oxidation, the higher the ketogenesis, the lower the chances of fat accumulation in the liver, thus reducing the incidence of steatosis. The initiation and progression of liver cancer begins with fatty liver and ensuing cirrhotic liver disease. As such, HMGCS2 functions as a tumor suppressor in this organ. In fact, since metabolic dysregulation (i.e., decreased fatty acid oxidation and ketogenesis; increased circulating fatty acids; and elevated cellular levels of fatty acids and triglycerides) is the hallmark of nonalcoholic fatty liver disease (NAFLD), the forerunner of hepatocellular carcinoma, NAFLD is now called MASLD (metabolic dysfunction-associated steatotic liver disease). Ketogenic diets, which increase HMGCS2 expression and promote ketogenesis in the liver, protect against liver cancer [143,156]. There is clear evidence for the protective role of HMGCS2 against liver cancer and for the efficacy of ketone bodies in suppressing cancer cell proliferation and migration, and hence tumor growth and metastasis [110,157,158,159]. Liver cancer is associated with marked downregulation of this protective enzyme, and the low expression correlates with vascular invasion, metastasis, and worse overall and disease-free survival. In this regard, it is relevant to note here that one of the negative regulators of HMGCS2 expression is the oncogene c-Myc [134,160]. HMGCS2 and ketone bodies not only protect against liver cancer but also sensitize liver cancer cells to the chemotherapeutic agent sorafenib [161,162].

The colon is another tissue with robust HMGCS2 expression, almost equal to the level of expression in the liver. Here too cancer is associated with marked downregulation of HMGCS2 expression [36,160,163,164]. In colon cancer cells, HMGCS2 suppresses differentiation and proliferation and also reprograms metabolic pathways. Cancer cells exhibit a unique set of reprogrammed metabolic pathways, including aerobic glycolysis, glutaminolysis, reductive carboxylation, increased one-carbon metabolism, improved antioxidant machinery, and enhanced synthesis of serine and glycine from glycolytic intermediates [165]. Decreased expression of HMGCS2 in colon cancer cells leads to suppression of fatty acid oxidation, increased dependence on glucose for energy, potentiation of fatty acid synthesis, and other metabolic phenotypes that are characteristic of highly proliferating cancer cells. Stated differently, high expression of HMGCS2 is not compatible with cell differentiation and proliferation. The impact of HMGCS2 expression level in colon cancer cells is not limited to its local effects within the cancer cell; it also influences the tumor microenvironment. It has been documented that in colon cancer tissue specimens, the expression level of HMGCS2 negatively correlates with microvessel density, indicating the impact of tumor-cell HMGCS2 on tumor angiogenesis [164]. In a co-culture system, knockdown of HMGCS2 in colon cancer cells enhances endothelial cell proliferation, branching, and tube formation, suggesting that decreased expression of HMGCS2 in cancer cells promotes tumor angiogenesis with tumor-cell secretion of some not yet identified factor(s) which influence the biology of endothelial cells in the tumor microenvironment. However, the role of HMGCS2 in colon cancer is not without controversy. Some studies have shown that HMGCS2 promotes migration, invasion, and metastasis of colon cancer cells and that the underlying mechanism is ketogenesis-independent and involves direct interaction of HMGCS2 protein with the nuclear receptor PPARα and resultant expression of the oncogene Src [166]. Whether these differences among the various studies in terms of whether or not HMGCS2 is a tumor suppressor or tumor promoter in the colon can be explained by the expression status of PPARα in the tumor tissue is yet to be investigated.

A similar situation exists with regard to the role of HMGCS2 in prostate cancer. Some studies have found evidence for a tumor-suppressive role [167], whereas others have found evidence for a tumor-promoting role [168,169]. In studies describing a tumor-promoting role, the expression of HMGCS2 is elevated in tumor cells, an upregulation that is mediated by tumor-associated fibroblasts [169]. The differences among these different studies may be related to the subtype of prostate cancer depending on whether the tumor is androgen-dependent or independent. When the cancer-associated fibroblasts induce HMGCS2 in tumor cells, the result is not ketogenesis but rather cholesterol and steroid synthesis. This scenario seems somewhat similar to the role of HMGCS2 in gonadal tissues, where the enzyme promotes the cholesterol-related mevalonate pathway instead of the ketone bodies-related ketogenic pathway [104,105]. Androgen-independent prostate cancer is generally associated with increased expression of the androgen receptor; therefore, it is likely that this subtype of prostate cancer has the biosynthetic machinery to generate cholesterol to feed into local synthesis of androgen.

In summary, a clearcut evidence for the tumor-suppressive role of HMGCS2 and ketogenesis exists only for liver cancer. In contrast, a tumor-suppressive role as well as a tumor-promoting role have been demonstrated for HMGCS2 in breast cancer and prostate cancer in different studies. The role of HMGCS2 in breast cancer and prostate cancer is complicated because of the dependence of these cancers on steroid hormones whose synthesis might be promoted by HMGCS2-mediated cholesterol synthesis. This could also explain the observed ability of HMGCS2 to induce resistance to tamoxifen in breast cancer and to promote the growth of castration-resistant prostate cancer. In addition, differences in tumor microenvironments and stromal cells (e.g., tumor-associated fibroblasts, tumor-associated macrophages and other immune cells) and their influence on the biology of tumor cells in different subtypes of breast cancer and prostate cancer also have to be considered in this context. In colon cancer, most studies have shown a tumor-suppressive function for HMGCS2.

### 7.4. Neurodegeneration

It has been known for decades that ketogenic diets are protective against various neurological diseases, including epilepsy, Alzheimer’s disease, and Parkinson’s disease [170,171,172,173]. Initially, the underlying mechanism for this protective effect was thought to be related solely to the ability of ketone bodies to serve as an alternative energy source for neurons when glucose availability or utilization is impaired. This may still be true for the efficacy of ketogenic diets to alleviate the symptoms of epilepsy and diabetes-associated neurological complications. But now that we know of the various other biological functions of ketone bodies, additional mechanisms need to be considered, including suppression of inflammation, blockade of microglia activation, improvement of mitochondrial morphology and function, reprogramming of neuronal metabolic pathways, changes in neurotransmitters, maintenance of antioxidant machinery, autophagic disposal of protein aggregates, and preservation of blood–brain barrier integrity. In this regard, the impact of ketone bodies on the biology of microglia merits special mention because of the overwhelming evidence for the role of microglial activation in the etiology of neurodegenerative diseases [174]. Activation of microglia involves polarization of these cells into the M1 proinflammatory phenotype, with suppressed ability to produce IL-10 and increased ability to produce IL-17. This initiates and potentiates inflammation in the brain. There is evidence for the ketone bodies in suppressing this microglial activation [175]. β-Hydroxybutyrate promotes polarization of microglial cells into the M2 anti-inflammatory phenotype, with consequent reversal in the M1-type generation of IL-10 and IL-17, thus leading to protection against inflammation. The underlying mechanism may involve activation of the immunosuppressive receptor GPR109A as well as improving mitochondrial function, with a resultant decrease in the production of reactive oxygen species.

## 8. HMGCS2 Deficiency

### 8.1. Loss-of-Function Mutations in HMGCS2

HMGCS2 deficiency (Online Mendelian Inheritance in Man ref. no. 605911) is a rare, autosomal recessive, inborn error of metabolism due to loss of function in the HMGCS2 protein arising from mutations in the gene coding for the enzyme. To date, only ~100 patients have been reported in the literature. With more than 40 different disease-causing variants of this protein, a wide variety of mutations have been reported in different ethnic groups throughout the world [176,177]. Loss of function in HMGCS2 variants arises from missense mutations with amino acid substitutions in the protein sequence, nonsense mutations with premature stop codon, as well as splicing and frame-shift mutations causing changes in the primary sequence [178,179,180,181,182,183,184,185,186,187,188]. It is interesting that so many different disease-causing mutations are found in this protein even though the number of patients is so few. Furthermore, the reported mutations are spread throughout the entire protein. There seems to be no obvious selection pressure for any particular mutation and there appears to be no mutational hotspot in the protein. Puisac et al. [179] carried out a detailed analysis of the structural implications of several mutations. A notable feature is that the disease-causing mutations are only rarely found at the catalytic site (amino acids 151–171) of the enzyme. Most mutations are located outside this site, raising an interesting question as to the molecular mechanism for the loss of functional activity of the enzyme due to these mutations. One plausible explanation is that the catalytic site just represents the location where the cysteine residue involved in the covalent binding of the intermediate during the catalytic cycle is present; other regions in the protein play an essential role in the maintenance of the structure of this catalytic site for effective initiation and completion of the reaction [179]. In many cases, the enzymatic activity of the disease-causing mutations has been investigated; the residual activity is ≤10% in a majority of cases. One notable exception, however, is the mutation R505Q, which retains ~70% of activity and yet causes the disease [179]. This may not necessarily mean that even the loss of 30% activity is sufficient to cause the disease because HMGCS2 deficiency is an autosomal recessive disorder and heterozygotes with one normal copy of the gene do not exhibit noticeable clinical symptoms. The enzyme activity is measured in an in vitro setting, and whether the findings are directly translatable to in vivo remains a question. Interestingly, despite the retention of ~70% catalytic activity, the patient harboring the mutation does show biochemical features indicative of decreased ketogenesis (e.g., reduced ketones in urine, dicarboxylic aciduria, hypoglycemia). Another point for consideration is that enzymatic activity is not the only biological function of the protein. It is already known that HMGCS2 can translocate into the nucleus and influence the transcriptional activity of the nuclear receptor PPARα [132,133]. The R505Q mutation does not interfere with enzyme activity to any marked extent, but the impact of this mutation on other non-enzyme-related functions of the protein remains unknown and deserves further investigation.

### 8.2. Clinical Consequences of HMGCS2 Deficiency

A recent review summarizes the clinical symptoms associated with homozygous HMGCS2 deficiency [188]. Onset of symptoms occurs as early as 3 months of age and most patients present with clinical episodes within 3 years of age. Both boys and girls are affected. If left untreated, the symptoms can potentially progress into coma. Generally, clinical symptoms begin to appear during fasting and/or prolonged illness. Most symptoms are nonspecific, and the biochemical findings also overlap with those found in other diseases, thus making diagnosis difficult. The generic symptoms include frequent infections, anorexia, vomiting, weakness, lethargy, and enteritis with diarrhea. Neurological complications indicative of encephalopathy include seizures and loss of consciousness. Biochemically, HMGCS2 deficiency leads to the blockade of ketogenesis and, therefore, blockade of fatty acid oxidation due to the backflow of metabolites in the pathway. This provides a mechanistic explanation for many of the clinical and biochemical features (Figure 5).

In these patients, plasma levels of ketone bodies fail to rise in response to fasting. Defective ketogenesis leads to the suppression of fatty acid oxidation, which causes increased fat deposition in the liver, leading to steatosis, hepatomegaly, and tissue damage. This forms the basis for hyperammonemia, hypofibrinogenemia, and elevated levels of triglycerides, low-density lipoprotein, and transaminases (alanine transaminase and aspartate transaminase) in circulation. Inability of the liver to generate ketone bodies leads to decreased ketone bodies in the blood and increased dependence on glucose for energy production in most tissues, thus explaining hypoketotic hypoglycemia seen in a majority of these patients. Defective fatty acid oxidation leads to elevated levels of acylcarnitines in the blood, which is expected because fatty acids enter mitochondria in the form of acylcarnitines for subsequent conversion to their CoA derivatives within the matrix prior to entry into the β-oxidation pathway. Concomitant with increased levels of acylcarnitines in circulation, free carnitine levels are decreased. Defective fatty acid oxidation forces partially metabolized fatty acids with shortened chain length into alternative biochemical routes, resulting in increased production of medium-chain dicarboxylic acids such as glutarate, which presents as dicarboxylic aciduria. On the same biochemical basis, medium-chain fatty acids with a hydroxyl group at the third carbon (e.g., 3-hydroxyoctanoic acid, 3-hydroxyhexanoic acid), which are intermediates during β-oxidation of long-chain fatty acids, are also present in urine at higher levels than normal. Treatment of patients with HMGCS2 deficiency primarily involves the management of metabolic crises by avoiding long-term fasting, the instillation of a low-fat diet, and the administration of intravenous glucose and bicarbonate during acute episodes of hypoglycemia and metabolic acidosis.

Most of the information currently available in the literature on the biologic consequences of defective ketogenesis in humans relates primarily to children affected with loss-of-function mutations in HMGCS2. We know almost nothing on the long-term consequences of this disorder in humans. A recent study using Hmgcs2-deficient mice sheds some light on this issue [189]. Defective ketogenesis caused by the absence of Hmgcs2 shortens lifespan in mice; however, this detrimental outcome is prevented by dietary supplementation of 1,3-betanediol, which is converted to β-hydroxybutyrate in the liver by the combined actions of alcohol dehydrogenase and aldehyde dehydrogenase without the involvement of Hmgcs2 [190]. Interestingly, supplementation of 1,3-butanediol in normal mice decreases life span, highlighting the conundrum in the long-term use of ketogenic diets under normal conditions. Obviously, provision of the ketone body precursor is beneficial when endogenous ketogenesis is impaired, but whether it represents good judgment to use such supplementations under normal conditions for long periods of time remains highly questionable.

## 9. Conclusions and Perspectives

The rise of ketone bodies in status from a simple alternative energy source to important signaling and regulatory molecules is phenomenal. These metabolites elicit a broad spectrum of biological effects that are generally beneficial and protective against a variety of diseases. The well-recognized health-promoting results of behavioral modifications such as regular exercise and intermittent fasting as well as dietary changes such as the use of ketogenic diets are at least partly due to the elevation of circulating ketone body levels. The clinical complications of ketoacidosis as a health crisis in diabetic patients or even in normal subjects with excessive use of high-fat/low-carbohydrate “keto” diets have painted a dark picture of ketone bodies as something undesirable. Ketone bodies are acids, and too much of these metabolites do precipitate a metabolic crisis in the form of acidosis. However, it has to be borne in mind that the body generates these metabolites by tapping into the substantial stores of fat in the body to protect critical organs from energy deficit when glucose is not available or cannot be used as an energy source. The biological significance of ketogenesis is evident from the clinical complications seen in patients with HMGCS2 deficiency. However, it is difficult to get a handle on the full scope of the importance of this biochemical pathway just based on the symptoms in these patients. This is because most of the patients with this disorder have been evaluated only as infants, and the symptoms associated with the disease can be easily explained based on one single function of ketone bodies, namely as an alternative energy source. But ketone bodies are definitely more than just energy substrates. They function as hormones in terms of their ability to activate specific cell-surface receptors and also as epigenetic modifiers with significant impact on the transcription of a broad spectrum of gene targets. Therefore, it is important to understand more about the potential long-term consequences of defective ketogenesis. HMGCS2 deficiency was discovered as a disease entity only about thirty years ago [191], and what we know about this disease is all about complications during the early stages of development. Hopefully, long-term follow-up of these patients into adulthood and beyond with a focus on neurodegeneration (e.g., Alzheimer’s disease), cancer (particularly liver cancer and colon cancer), inflammatory diseases (e.g., Crohn’s disease and ulcerative colitis), and metabolic diseases (e.g., diabetes) would shed more light on the biological functions of ketone bodies as a whole.

## Figures and Tables

**Figure 1 biomolecules-15-00580-f001:**
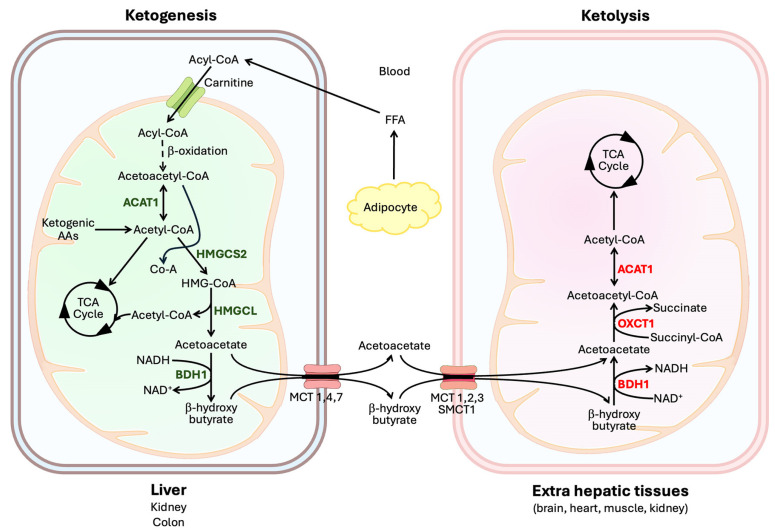
Reactions involved in ketogenesis and ketone utilization. ACAT1, acetyl CoA acetyltransferase 1; HMGCS2, 3-hydroxy-3-methylglutaryl-CoA synthase 2; HMGCL, 3-hydroxy-3-methylglutaryl-CoA lyase; BDH1, β-hydroxybutyrate dehydrogenase 1; OXCT1, 3-oxoacid CoA transferase 1; FFA, free fatty acids; AAs, amino acids; MCT, monocarboxylate transporter; SMCT1, sodium-coupled monocarboxylate transporter 1; TCA, tricarboxylic acid.

**Figure 2 biomolecules-15-00580-f002:**
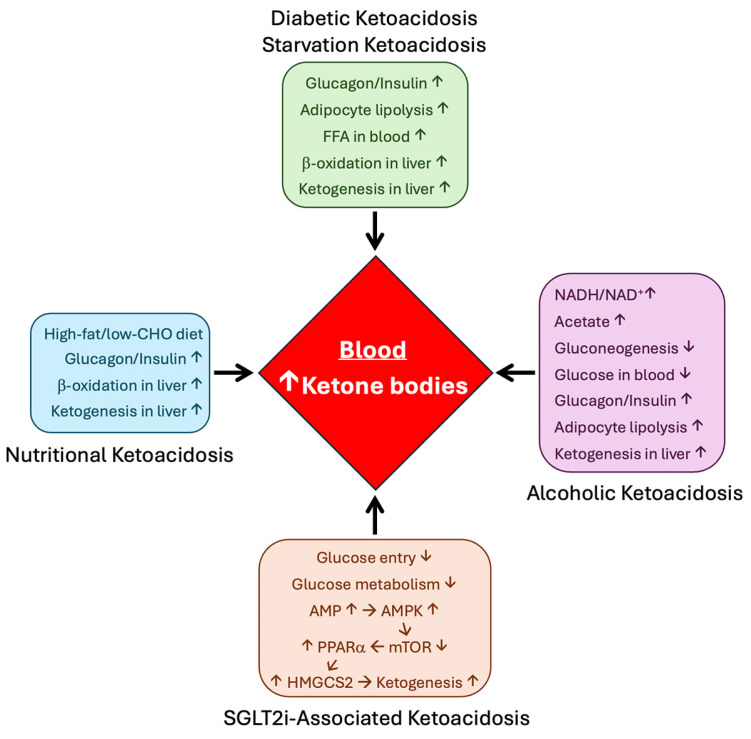
Biochemical mechanisms underlying ketoacidosis under various physiological, pathologi-cal and pharmacological conditions. FFA, free fatty acids; CHO, carbohydrate; HMGCS2, 3-hydroxy-3-methylglutaryl-CoA 2; AMPK, AMP-activated kinase; mTOR, mechanistic target of rapamycin; PPARα, peroxisome proliferator-activated receptor α; SGLT2i, sodium-coupled glucose transporter 2 inhibitors.

**Figure 3 biomolecules-15-00580-f003:**
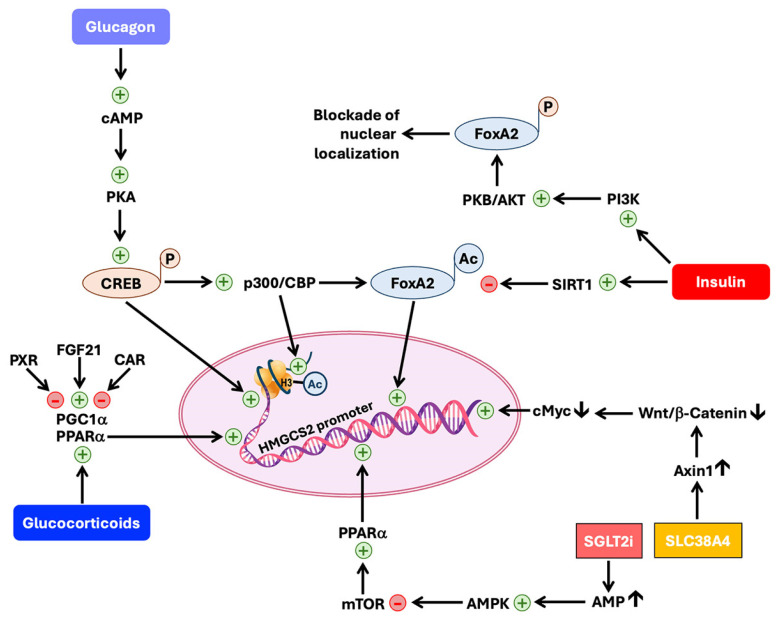
Regulation of HMGCS2 at the transcriptional level. PKA, protein kinase A; CREB, cAMP-responsive element binding protein; CBP, CREB binding protein; FoxA2, forkhead box protein A2; PKB, protein kinase B; PI3K, phosphatidylinositol-3-kinase; PXR, pregnane X receptor; CAR, constitutive androstane receptor; PPARα, peroxisome proliferator-activated receptor α; PGC1α, peroxi-some proliferator-activated receptor gamma coactivator 1α; FGF21, fibroblast growth factor 21; P, phosphorylated; Ac, acetylated; H3, histone 3. The positions of the individual cis-elements in the HMGCS2 gene promoter that are responsible for the binding of the indicated transcription factors are simply schematic and do not represent the exact positions in relation to one another.

**Figure 4 biomolecules-15-00580-f004:**
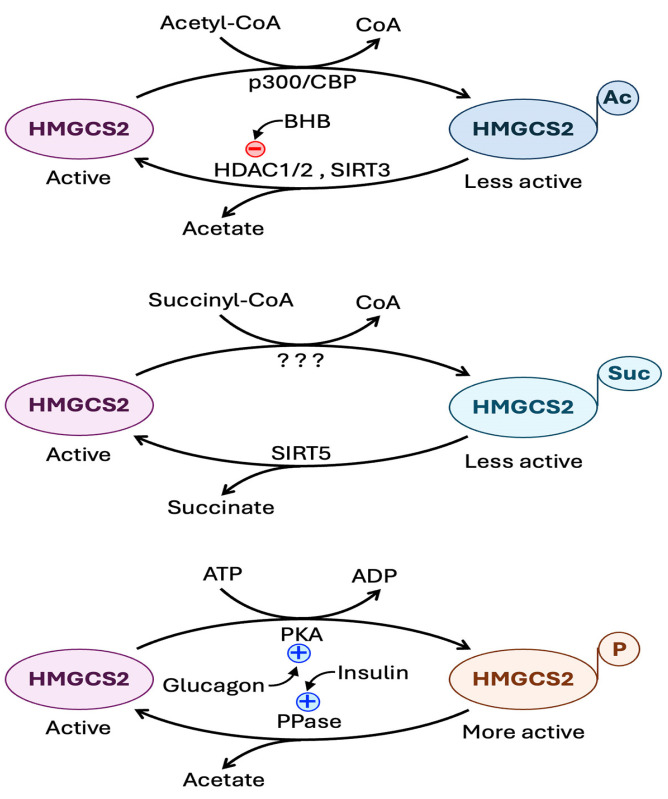
Regulation of the catalytic activity of HMGCS2 by post-translational modifications. HMGCS2, 3-hydroxy-3-methylglutaryl-CoA 2; Ac, acetylated; Suc, succinylated; P, phosphorylated; BHB, β-hydroxybutyrate; HDAC, histone deacetylase; PKA, protein kinase A; PPase, protein phosphatase 1; CBP, CREB binding protein.

**Figure 5 biomolecules-15-00580-f005:**
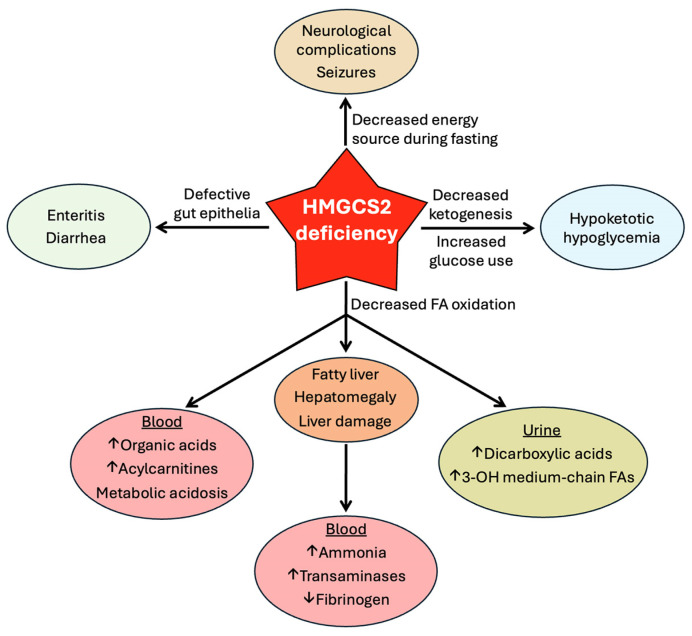
Clinical symptoms of HMGCS2 deficiency and their molecular basis. HMGCS2, 3-hydroxy-3-methylglutaryl-CoA synthase 2; FAs, fatty acids; ↑ and ↓ indicate an ‘increase’ or ‘decrease’, respectively.

**Table 1 biomolecules-15-00580-t001:** Expression pattern of HMGCS1 and HMGCS2 at the protein level in human tissues. The protein levels were assessed by immunohistochemistry with HMGCS1- specific and HMGCS2-specific antibodies on histological sections of normal tissues (The Human Protein Atlas database [103]).

Expression Level	HMGCS1	HMGCS2
High	Esophagus Stomach Small intestine Large intestine Liver Testis	Urinary bladder Gallbladder Small intestine Large intestine Liver Kidney
Moderate	Nasopharynx Bronchus Lung Oral mucosa Gallbladder Pancreas Kidney Urinary bladder Epididymis Seminal vesicle Prostate Vagina Endometrium Cervix Placenta Skin	Stomach Testis Mammary gland

## Data Availability

Not applicable.

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
