# Peer review of "Not Just an Alternative Energy Source: Diverse Biological Functions of Ketone Bodies and Relevance of HMGCS2 to Health and Disease"

_biomolecules, 2025, doi:10.3390/biom15040580_

Round 1
Reviewer 1 Report
Comments and Suggestions for Authors
Attached

Author Response
Comment 1:
I do not see the relevance to the manuscript about the differences between type 1 and type 2 diabetes (lines 62 to 70). And the same about the 3-hydroxy-3-methylbutyrate (HMB) (lines 103 to 107).
Response: These portions have now been deleted in the revised version (page 2 and page 3).
Comment 2:
Reduce and simplify the section 3 “Ketogenesis in Different Organs”, especially the subsection regarding the liver. I am aware of the importance of liver in the ketogenesis pathway but in my opinion, most of the information are in physiology specialized text books.
Response: We have made our best efforts to shorten this section on ketogenesis in liver in the revised version (page 4). However, we did not cut the portion drastically because we believe that the section contains important information on linking four different metabolic pathways, namely ketogenesis, beta-oxidation, gluconeogenesis, and urea genesis, in the liver. This kind of explanation for linking all four pathways is not found in text books, but I provide this logical explanation to medical students when teaching biochemistry. I tried to do the same in this review because I think it would be very useful for readers.
Comment 3:
A ketogenesis pathway has been described to be produced in the cytosol of the cells, because a cytosolic HMG-CoA lyase enzyme was described in 2012 by Arnedo et al and Montgomery et al. Why do not even mention this alternative ketone body synthesis pathway?
Response: We have now added this new information as recommended by the reviewer on page 16 of the revised version. This includes two new references (109 and 110).
Comment 4:
Could you please simplify the section “Regulation at the level of transcription of HMGCS2 gene”? It is too long and sometimes not easy to follow.
Response: We have tried our best to shorten and simplify this section to make it easier to read and understand (pages 17-19 in the revised version).
Comment 5:
The role of HMGCS2 in cancer is complex and depends on the type of cancer as you already mention it. Because of that, in my opinion, you should reduce or simplify this section. A table with the different types of cancer and the findings regarding the HMGCS2 enzyme will help.
Response: We understand the reviewer's concern regarding the confounding data in the literature on the role of HMGCS2 in different cancers. The reviewer asked for a Table describing this information. Instead of a Table, we have opted to include a new paragraph (pages 22-23) summarizing the information. This serves the same purpose as a Table would do. We hope that this is okay with the reviewer and the editor.
Reviewer 2 Report
Comments and Suggestions for Authors
Manuscript: Not just an alternative energy source: Diverse biological functions of ketone bodies and relevance of HMGCS2 to health and disease
1. Clarity and consistency in the analysis of the role of HMGCS2 in tumors
The manuscript highlights both the potential tumor suppressor and pro-oncogenic role of HMGCS2, but lacks a coherent discussion of the conditions under which this enzyme can promote or inhibit tumor progression. I suggest:
Include a summary paragraph summarizing the main mechanisms determining the dual function of HMGCS2.
Expand the discussion on tumor microenvironment-dependent effects, such as the influence of fibroblasts and local metabolic conditions.
Add a comparative table illustrating the pro- and anti-oncogenic effects of HMGCS2 in different cancer types.
Authors may consider the following paper to improve the introduction: Polito et al., The Ketogenic Diet and Neuroinflammation: The Action of Beta-Hydroxybutyrate in a Microglial Cell Line, International Journal of Molecular Sciences, 2023, 24(4), 3102
2. Further explore the long-term effects of HMGCS2 deficiency
The manuscript discusses HMGCS2 genetic deficiency in a limited way, focusing on acute symptoms in neonates, but without exploring the long-term effects. It would be helpful to:
Discuss the potential impact on brain metabolism and neurodegenerative diseases.
Consider whether HMGCS2 deficiency could influence adult metabolic diseases, such as diabetes or neuropsychiatric disorders.
Include references to longitudinal studies or animal models that have explored disease progression.
3. Better contextualize the therapeutic use of ketone bodies
The manuscript mentions the potential therapeutic applications of ketone bodies, but without exploring the possible clinical risks and limitations. To improve this section, I suggest:
Discuss the balance of benefits and risks associated with ketogenic diets, especially for patients with pre-existing conditions (diabetes, cardiovascular disorders).
Provide data on the long-term tolerance and safety of interventions that increase ketogenesis.
Consider whether there are strategies to selectively modulate ketogenesis based on the metabolic needs of patients.
4. Further investigate the regulation of ketogenesis in pathological conditions
The manuscript discusses the regulation of HMGCS2 in physiological conditions, but lacks a detailed discussion of how it is altered in specific diseases. It would be helpful to:
Expand the discussion on the role of insulin resistance and glucagon in aberrant ketogenesis, especially in diabetes and chronic inflammatory diseases.
Consider the interaction between HMGCS2 and key metabolic transcription factors, such as PPARα and FGF21, that regulate ketogenesis.
Include an analysis of the differences between hepatic and renal ketogenesis under conditions of metabolic stress.
5. Improve the section on diagnosis and treatment of HMGCS2 dysfunction
The diagnosis of HMGCS2 deficiency and ketogenesis abnormalities is treated generically. For greater clinical utility, I suggest:
Deepen the diagnostic biomarkers for HMGCS2 alterations. Are there specific tests to assess their activity?
Discuss possible experimental therapeutic strategies, such as the use of ketone body precursors or epigenetic regulators.
Consider whether there are possible personalized medicine approaches to modulate ketogenesis in patients with metabolic dysfunction.
Author Response
Comment 1:
Clarity and consistency in the analysis of the role of HMGCS2 in tumors. The manuscript highlights both the potential tumor suppressor and pro-oncogenic role of HMGCS2, but lacks a coherent discussion of the conditions under which this enzyme can promote or inhibit tumor progression.
Response: We tried to summarize this information in a new paragraph describing the potential involvement of tumor microenvironment and stratal cells and also the varied functions of HMGCS2 in ketogenesis and in cholesterologenesis to provide a molecular basis for the tumor suppressive and tumor promotive roles of the enzyme in different cancers. This paragraph appears on pages 22-23 in the revised version. The reviewer also suggested inclusion of the data from a new published report to emphasize the role of microglial activation in neuroinflammation. We have now added this new information in the revised version on page 23 under the subheading "neurodegeneration" and cited the reference (179).
Comment 2:
Further explore the long-term effects of HMGCS2 deficiency. The manuscript discusses HMGCS2 genetic deficiency in a limited way, focusing on acute symptoms in neonates, but without exploring the long-term effects.
Response: There is no information available in the literature on the long-term effects of HMGCS2 deficiency. We mentioned this in our original version. However, there is some data on the long-term effects of Hmgcs2 deletion in mice. We have now added this new information in the revised version as a separate paragraph (pages 25-26).
Comment 3:
Better contextualize the therapeutic use of ketone bodies. The manuscript mentions the potential therapeutic applications of ketone bodies, but without exploring the possible clinical risks and limitations.
Response: Ketone bodies have been tried as a treatment only in mice so far to counter the symptoms associated with Hmgcs2 deficiency. We have now added this new information in the revised version (pages 25-26; Ref. 193). This paper also talks about the potential detrimental effects of dietary supplementation of ketone bodies on a long-term basis under normal conditions. This point is also mentioned in the revised version. We did not go into the use of ketogenic diets under normal conditions and in disease states because we do not believe that this particular aspect is the central focus of our review. Hope that the reviewer would concur with our rationale.
Comment 4:
Further investigate the regulation of ketogenesis in pathological conditions. The manuscript discusses the regulation of HMGCS2 in physiological conditions, but lacks a detailed discussion of how it is altered in specific diseases.
Response: We do not know how to respond to this comment. The manuscript details regulation of the enzyme (gene expression and enzymatic activity) under normal conditions. Under disease states that are associated with alterations in the relevant signaling pathways, the expression and activity of the enzyme will be altered, but this can be easily inferred based on the involvement of each of the signaling pathways in the process. This information is already included in the manuscript.
Comment 5:
Improve the section on diagnosis and treatment of HMGCS2 dysfunction. The diagnosis of HMGCS2 deficiency and ketogenesis abnormalities is treated generically.
Response:
The clinical symptoms of HMGCS2 deficiency overlap with symptoms of several other diseases. Therefore, it is difficult to differentially diagnose this particular disorder. Furthermore, only about 100 patients have been identified and studies thus far and therefore the differential diagnosis for this disorder is constantly evolving. There are no specific biomarkers for this disorder. Dicarboxylic acuduria that is seen in this disorder is also seen in other conditions. This is the reason for the lack of specific details on this disorder in the literature and in our review.